# A Review on the Green Synthesis of Benzimidazole Derivatives and Their Pharmacological Activities

Monica Nardi [1,*], Natividad Carolina Herrera Cano [2,*], Svilen Simeonov [3,4,*], Renata Bence [2], Atanas Kurutos [3], Rosa Scarpelli [1], Daniel Wunderlin [2] and Antonio Procopio [1]

[1] Dipartimento di Scienze della Salute, Università Magna Græcia, Viale Europa, Germaneto, 88100 Catanzaro, Italy

[2] ICYTAC, CONICET and Departamento Química Orgánica, Facultad de Ciencias Químicas, Universidad Nacional de Córdoba, Ciudad Universitaria, Bv. Juan Filloy s/n, Córdoba 5000, Argentina

[3] Institute of Organic Chemistry with Centre of Phytochemistry, Bulgarian Academy of Sciences, bl. 9, Acad. G. Bonchev str., 1113 Sofia, Bulgaria

[4] Research Institute for Medicines (iMed.ULisboa), Faculty of Pharmacy, Universidade de Lisboa, Av. Prof. Gama Pinto, 1649-003 Lisbon, Portugal

\* Correspondence: monica.nardi@unicz.it (M.N.); nherrerac@unc.edu.ar (N.C.H.C.); svilen.simeonov@orgchm.bas.bg (S.S.); Tel.: +39-0961-3694116 (M.N.)

**Abstract:** Benzimidazoles and their derivatives play an extraordinarily significant role as therapeutic agents, e.g., antiulcer, analgesic, and anthelmintic drugs. The organic synthesis of benzimidazoles and derivatives to obtain active pharmacological compounds represents an important research area in organic chemistry. The use of non-environmental organic compounds and application high energy synthetic methods, the production of waste, and the application of conventional toxic processes are a problem for the pharmaceutical industry and for these important drugs' synthesis. The substituted benzimidazoles are summarized in this review to provide insight about their organic synthesis using ecofriendly methods, as well as their pharmacological activities.

**Keywords:** benzimidazole; green chemistry; pharmacological activity

## 1. Introduction

Since Woolley proposed in 1944 that benzimidazole may behave similarly to purines, stimulating various biological reactions, the therapeutic potential of benzimidazole nucleus has been known [1]. Some years later, Brink determined that 5,6- dimethylbenzimidazole is a vitamin B12 breakdown product and discovered that several of its derivatives had action similar to that of vitamin B12 [2,3]. Due to its presence in a variety of bioactive substances, including antihypertensives, anti-inflammatories, antivirals, analgesics, anticancer, proton pump inhibitors, anticonvulsants, antifungals, anticoagulants, antihistaminics, antiparasitics, and antiulcers, the development of the benzimidazole core has emerged over the recent years [4–11]. Thus, research on the synthesis of bioactive molecules from benzimidazole has significantly been accelerated over the last decade.

Literature study shows that the different derivatives of benzimidazole have been synthesized for their pharmacological activities. The present review fits into this framework by discussing the literature existing in recent years on strategies for the reduction and replacement of hazardous solvents affording the preparation of benzimidazoles. Additionally, it discloses numerous benzimidazole derivatives with different pharmacological activities based on the substitution model around the nucleus.

## 2. Eco-Friendly Synthesis of Benzimidazoles and Derivatives

Due to synthetic importance and various bioactivities showed by benzimidazoles and their derivatives, a major effort has gone into generating libraries of these compounds.

In the primary nineties, numerous benzimidazole derivatives were synthesized with substitution of fluorine, propylene, tetrahydroquinoline, and cyclized molecule, obtaining compounds with superior stability and good biological activity [12,13]. Synthetic benzimidazole products containing electron donating group have been demonstrated to be effective antiulcer drugs [14,15], for example the Omeoprazole. Instead, other benzimidazole derivatives have shown healing activity in diseases such as ischemia-reperfusion injury or hypertension [16].

The initial synthetic methods described in the literature have shown o-phenylenediamine reacting with carboxylic acids or their derivatives [17,18].

Mono acyl derivatives of *o*-phenylenediamine are converted into the corresponding benzimidazole under a temperature above the melting point of the starting compounds in an atmosphere of nitrogen to prevent oxidation [19].

Subsequently, synthetic methods replaced carboxylic acids with aldehydes obtaining 2-substituted and 1,2-substituted benzimidazoles. These synthetic procedures, however, showed different complications for long reaction times, under drastic conditions, and using toxic solvents [20–22]. Furthermore, waste production and non-recoverable and poorly green and selective catalysts are often used.

The use of toxic solvents and the formation of a large amount of industrial waste are grave problems for the environment and human health. Recently, green chemistry principles have inspired the activities of pharmaceutical industries, suggesting use of environmental solvents [23,24], reducing waste production by selective reaction methods and recyclable reagents [25–40]. This circumstance has directed academia and industry to make substantial efforts towards the development of alternative synthetic routes.

## 2.1. Use of Catalysis for the Synthesis of Benzimidazoles

Since the development of innovative synthetic processes to obtain potential drug molecules has become a significant research field, the pursuance of more suitable and practical synthetic methods for benzimidazoles remains an active research area. Furthermore, the use of catalysts has become very important.

The use of Lewis acids as efficient catalysts in various transformations proved to be a greener alternative method for the synthesis of benzimidazole derivatives. A facile synthesis of benzimidazole derivatives using *o*-phenylenediamines and orthoesters at room temperature is the first example of the synthesis of these compounds (Scheme 1) [41].

R = H, Me, Cl, $NO_2$
$R_1$ = H, Me, Et, $CH_3(CH_2)_3$
$R_2$ = Me, Et

**Scheme 1.** $ZrOCl_2$. $8H_2O$, $TiCl_4$, $SnCl_4$, $5H_2O$, and $HfCl_4$ have been proved to be highly effective catalysts. In contrast, other metal salts such as $Mg(ClO_4)_2$, $Bi(NO_3)_5$, $5H_2O$, $CuCl_2$, $2H_2O$, $CoCl_2$, $6H_2O$, $ZnCl_2$, $Zn(ClO_4)_2$, $NiCl_2.6H_2O$, LiCl, $LiBr.H_2O$, $CdCl_2$, $2.5H_2O$, $Ti(SO_4)_2$, $NH_4Ce(NO_3)_2$, $SiCl_4$, $BF_3.Et_2O$, $BCl_3$, and $Zr(SO_4)_2.4H_2O$ were either less active or inert as a catalyst. In the absence of a catalyst, the reaction did not yield any product. The reaction proceeded efficiently in the presence of $ZrCl_4$ (10 mol%) and anhydrous EtOH at room temperature, with excellent yields.

Synthesis of disubstituted benzimidazole includes the use of catalysts such as TFE/HFIP, 6 CuI/L-proline [42], TMS [43], amberlite IR-120 [44], $SiO_2/ZnCl_2$ [45], Dowex-50 W [46], SDS micelles [47], silica sulfuric acid [48], $FePO_4$ [49], CAN [50], $Cu(NO_3)_2.3H_2O$ [51], and $FeCl_3/Al_2O_3$ [52]. In many cases, the reaction shows poor selectivity in terms of N-1 substitution, which results in the formation of a mixture of 1,2-disubstituted and 2-substituted benzimidazoles. Furthermore, major drawbacks of present protocols include the use of expensive reagents, long reaction times, and the use of hazardous organic solvents. Regarding the latter, substituted benzimidazoles have been synthesized from *o*-phenylenediamine

and ethyl a-cyanocinnamate by transfer-hydrogenation process. Kappor et al. proposed a highly efficient and metal-free transfer-hydrogenation process from in situ generated benzimidazolines to activated olefins. The reaction performed under solvent-free and catalyst-free conditions. heating ethyl a-cyanocinnamate with o-phenylenediamine in equimolar quantities at 100 °C. The starting material was found to vanish with the appearance of three new components. The most polar compound was isolated by precipitation and the remaining compounds were purified to obtain the corresponding benzimidazole (Scheme 2) [53].

**Scheme 2.** Transfer hydrogenation of electronically depleted olefins to obtain benzimidazole.

Synthesis of benzimidazole derivatives has been performed using arylaldehydes or arylmethylene-malononitriles as starting material under solvent and catalyst-free conditions, with silica gel used as a means of absorbing the starting materials [54]. The reaction was carried out by intermittent grinding or by a microwave-assisted technique (Scheme 3).

**Scheme 3.** Synthesis of benzimidazole derivatives by silica support.

Lanthanide triflates have been very successful in their application in the benzimidazoles catalytic synthesis as Lewis acid catalysts. $Zn(OTf)_2$ has been used to synthesize novel benzimidazole-linked triazole derivatives [55]. In addition to benzimidazoles, triazoles also exhibit various biological activities and are widely employed as pharmaceuticals and agrochemicals. In view of the biological importance of benzimidazole and 1,2,3-triazoles, to know the combined effect of the two moieties, it was considered worthwhile to synthesize certain new chemical products having benzimidazole and 1,2,3-triazole pharmacophores in a single molecule. The reaction performed by treatment of 2-(4-azidophenyl)-1H-benzo[d]imidazole (6) with different types of terminal alkynes in t-BuOH/$H_2O$, sodium ascorbate, and $Zn(OTf)_2$.

The use of $Er(OTf)_3$ as a commercially available and easily recyclable catalyst promoted the synthesis of 1,2-disubstituted benzimidazoles [56]. Additionally, 2-substituted benzimidazoles were selectively obtained in high yield and short reaction times in water as solvent at 1–2 °C or at 80 °C (for electron-deficient aldehydes) (Scheme 4).

**Scheme 4.** Synthesis of benzimidazole derivatives using $Er(OTf)_3$ as catalyst.

Next, catalytic applications of nanoporous materials in chemical synthesis have been highly successful. The use of zeolite as a hierarchical nanoporous material for the synthesis of benzimidazoles by the condensation of 1,2-phenylenediamine with aromatic aldehydes was investigated. The advantages of this procedure were high chemoselectivity and

shorter reaction time, but the procedure was performed in a toxic solvent as acetonitrile. While numerous research studies have been conducted on the use of non-toxic solvents, adopting methods based on solvent-free or solid-state reaction conditions are also effective to reduce pollution. In this context, solid Lewis's acid catalysts are usually used [57,58]. In the last years, the use of heterogeneous catalysts in solvent-free, microwave-assisted reactions has been mainly important for industrial production. In this regard, Bonacci et al. found it useful to use this heterogeneous catalyst for the synthesis of cyclopentenone derivatives from furfural [59] and benzimidazole derivatives [60]. The reaction to obtain benzimidazoles is performed under MW irradiation and in solvent-free conditions (Table 1). The reaction to optimize the conditions takes place between *o*-PDA (*o*-phenylenediamine) and benzaldehyde in a 1:1 or 1:2 molar ratio at different temperature and different wt (%) of MK10.

**Table 1.** Synthesis of benzimidazoles using MK10.

| Entry | MK10 wt (%) [a] | Molar Ratio *o*-PDA:Benzaldehyde | Temp (°C) | Time (min) | Conversion (%) [b] | Selectivity (%) [c] |
|---|---|---|---|---|---|---|
| 1 | 10 | 1:1 | rt | 120 | 19.3 | 12.0 |
| 2 | 10 | 1:2 | rt | 120 | 20.9 | 53.0 |
| 3 | 10 | 1:2 | 60 | 120 | 79.6 | 65.1 |
| 4 | 10 | 1:1 | 80 | 120 | 80.9 | 33.3 |
| 5 | 10 | 1:1 | 100 | 60 | 99.9 | 38.3 |
| 6 | 10 | 1:2 | 100 | 60 | 99.9 | 75.0 |
| 7 | - | 1:2 | 100 | 90 | 45.0 | 49.0 |
| 8 [d] | 20 | 1:1 | 60 | 5 | 99.9 | 18.2 |
| **9 [d]** | **20** | 1:2 | **60** | **5** | **99.9** | **98.5** |

[a] wt% with respect to amine. [b] Percent conversion of the *o*-PDA calculated from GC/MS data. [c] Percent yield calculated from GC/MS data of the corresponding disubstituted benzimidazole derivative. By-product obtained is constituted by mono substituted benzimidazole. [d] MW-assisted reactions. The reactions were performed on a Synthos 3000 instrument from Anton Paar equipped with a 4 × 24 MG5 rotor.

The complete conversion of *o*-phenylendiamine occurred using 20 wt% of MK10. The reaction was performed at 60 °C under MW irradiation. Furthermore, 2-phenyl-benzimidazole has been the principal product (81.2% yield). Using 2 mmol of benzaldehyde, 1-benzyl-2-phenyl-benzimidazole in 98.5% yield was obtained (Table 1, entry 9).

The essential benefit that is obtained in sustainability by using a heterogeneous catalyst is the catalyst recycling. To show this, after the complete conversion of the amine into the benzimidazole derivative, the final reaction mixture was treated with ethyl acetate to recover the MK10 by filtration, suitably washed, and dried. The recovered catalyst was used for the next run, adding fresh reagents following the procedures optimized (Figure 1).

Considering the excellent activity of Er (III) in Lewis acid catalyzed reactions under microwave irradiation [35,57,61,62] for the synthesis of benzimidazoles [61] and benzodiazepine [63,64] derivatives, the development of an original and environmental method MW-assisted for the synthesis of different substituted benzimidazoles is performed. The synthetic system does not involve the use of solvents, but involves the use of only 1% Er(OTf)$_3$ as a Lewis's acid catalyst for the synthesis of benzimidazole derivatives [65]. The use of only reagents to obtain desired products (91–99% yield) in faster reaction times (5 min) represents a valid synthetic process (Figure 2).

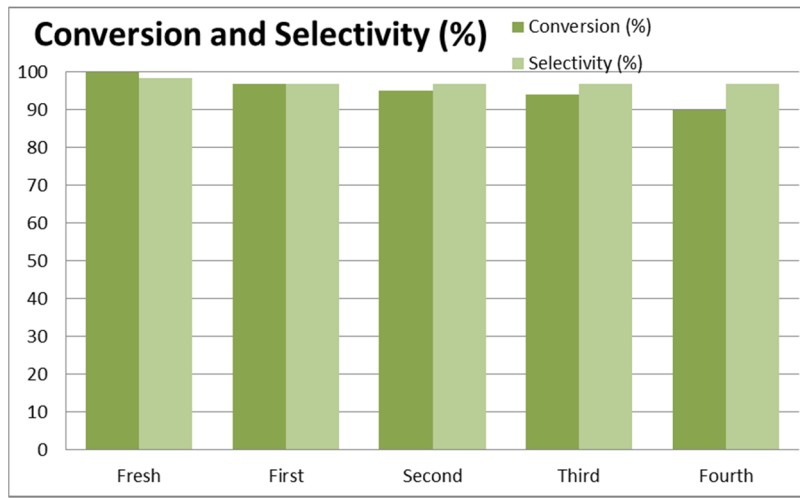

**Figure 1.** Cycling performing of MK10 in synthesis of 1-benzyl-2-phenyl-benzimidazole using the optimal reaction conditions.

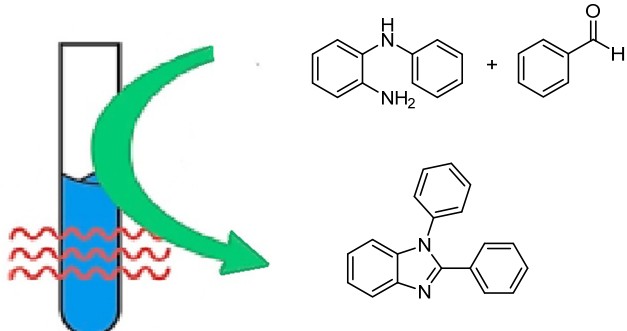

**Figure 2.** The environmental synthesis of 1,2-disubstituted benzimidazoles by microwave irradiation and $Er(OTf)_3$.

Recently, a very efficient copper (II) salt (20 mol%) catalyzed oxidative strategy for the amination of methyl arene using aniline and $TMSN_3$ as start material has been reported for the synthesis of functionalized benzimidazoles [66].

The proposed synthetic method requires the use of high temperatures (800 °C) and a non-recyclable external oxidant. Bhalla et al. [67] prepared PANI@Au:CuO nanocomposites (NCS), as recyclable photocatalyst for the synthesis of 2-arylbenzimidazole derivatives via sequential amination, azidation, and annulation reactions. The PANI-5@Au:CuO NCS was prepared using supramolecular assemblies of pentacenequinone derivative (Figure 3) as reactors and reducing agents. The reaction was performed using visible light as energy source over thermal conditions and in the absence of external oxidants/additives. Substituted methyl benzene, aniline, and $TMSN_3$ utilizing PANI-5@Au:CuO (1:1) NCS as photocatalyst gave the benzimidazole derivative (Scheme 5).

**Figure 3.** Pentacenequinone derivative.

**Scheme 5.** Sequential amination/azidation/annulation reactions of toluene derivatives bearing methyl groups at different positions with aniline and TMSN$_3$ utilizing PANI-5@Au:CuO (1:1) NCS as photocatalyst.

The in situ generated polyaniline species from pentacenequinone derivative acts as an oxidant to obtain the benzimidazoles derivative. This catalytic system does not require the use of elevated temperature and it can be recyclable.

*2.2. Synthesis of Privileged Benzimidazole Scaffolds Using Green Solvent*

In the last years, the most essential pharmaceutical industries have been inspired by green chemistry principles. The use of biomass derivatives, reduction of toxic solvents, reductions in waste production, and eco-friendly organic synthetic methods [68] have been introduced. In this regard, given the large amount of hazardous conventional solvents used by the pharmaceutical industry, most of the studies are currently converging on the use of more eco-friendly alternatives. The use of water or ionic liquids (ILs) as green media and/or the use of organometallic catalysts have been developed.

The synthesis of 2-substituted benzimidazole derivatives with equimolar amounts of aromatic aldehydes and *o*-phenylenediamine under microwave irradiation is investigated using [BMIM]HSO$_4$ as ionic liquid. The same method was also reported for the synthesis of 1,2-disubstituted benzimidazole derivatives by using 2-molar amounts of aromatic aldehydes in high yield [69]. One of the salient features of this method is that electron withdrawing as well as electron donating groups substitute aromatic aldehydes, giving excellent yields and purity.

It must be considered that ILs are toxic and dangerous to the environment [70]. Furthermore, their organic synthesis and purification are often costly and time-consuming [71]. Deep eutectic solvents (DES) are new green solvents with massive applicability in all areas of the chemical industry [72].

DESs are like ionic liquids from physical point of view, but quite different regarding the chemical character. Ionic liquids are composed by cations and anions, while DES are generally a combination of two or more components, with at least one hydrogen bond acceptor (HBA) and one hydrogen bond donor (HBD). These components (two or more natural compounds) interact between themselves by hydrogen bonding, behaving as one single entity. Because the production of these important solvents relies exclusively on the physical mixture of natural compounds, their production has practically no effect on the environment. These green solvents are also low-cost alternatives to most common solvents [73].

Recently, seven deep eutectic solvents were reported with the study of their physico-chemical properties. They are given in Table 2 and were used as catalysts for the synthesis of benzimidazole derivatives [74].

Initially, the reaction was conducted between benzaldehyde and o-phenylenediamine at 1:1 ratio in the absence of any catalyst, at room temperature, and no product was formed even after 6 h. Later, the reaction was repeated in the presence of the seven DESs to identify the better catalyst for the synthesis of benzimidazole derivatives, at room temperature. Among the seven DESs, excellent yield (95%) was observed in the presence of DES 1 (a combination of ZrOCl$_2$.8H$_2$O and urea at 1:5 ratio).

**Table 2.** Prepared DESs with their properties.

| Entry | DES | Molar Ratio | Abbreviation | Viscosity (cP) | pH | Catalytic Activity |
|-------|-----|-------------|--------------|----------------|-----|--------------------|
| 1 | ZrOCl$_2$.8H$_2$O: Urea | 1:5 | DES 1 | 52.40 | 2.20 | Higher yield |
| 2 | ZrOCl$_2$.8H$_2$O: Ethylene glycol | 1:2 | DES 2 | 149.50 | 6.99 | No reaction |
| 3 | ZrOCl$_2$.8H$_2$O: Glycerol | 1:2 | DES 3 | 532.73 | 6.99 | No reaction |
| 4 | CeCl$_3$.7H$_2$O: Urea | 1:5 | DES 4 | 195.39 | 4.71 | No reaction |
| 5 | CeCl$_3$.7H$_2$O: Ethylene glycol | 1:2 | DES 5 | 290.40 | 4.44 | No reaction |
| 6 | CeCl$_3$.7H$_2$O: Glycerol | 1:2 | DES 6 | 868.73 | 4.27 | No reaction |
| 7 | CeCl$_3$.7H$_2$O: Lactic acid | 1:4 | DES 7 | 564.64 | 2.02 | No reaction |

Viscosity was measured using Ostwald viscometer at room temperature with respect to water. pH measurements were conducted using ELICO pH meter which was pre-calibrated with buffer solutions of pH 4.0 and 7.0.

The remaining DESs showed no conversion of *o*-phenylenediamine to the corresponding benzimidazole derivative (Scheme 6).

**Scheme 6.** Formation of benzimidazole in the presence of DESs under mild conditions.

The model reaction was repeated to optimize the amount of DES 1 and reaction time (Table 3). At first, the reaction was conducted using benzaldehyde and *o*-phenylenediamine as starting material at 1:1 ratio in the presence of 0.2 mmol of DES 1, but low yield was observed. The reaction was repeated using different amounts of DES 1 and good yield was observed using 0.5 mmol of DES 1 and subsequently an increase in the amount of DES did not show an increase in the yield of the product.

**Table 3.** Optimization of the reaction conditions.

| Entry | DES 1 | | Time (min) | Isolated Yield (%) |
|-------|-------|--------|------------|--------------------|
| | (g) | (mmol) | | |
| 1 | 0.124 | 0.2 | 60 | 65 |
| 2 | 0.186 | 0.3 | 60 | 70 |
| 3 | 0.248 | 0.4 | 60 | 76 |
| 4 | 0.310 | 0.5 | 60 | 97 |
| 5 | 0.373 | 0.6 | 60 | 96 |
| 6 | 0.310 | 0.5 | 30 | 97 |
| 7 | 0.310 | 0.5 | 10 | 97 |

Reaction conditions: Benzaldehyde (1 mmol), *o*-PDA (1 mmol), DES 1, 30 °C.

In the last years, particular interest has been shown in the synthesis of DESs, in which one of the components of DES itself is the reactant to be converted into the reaction product. A new synthetic route to benzimidazole derivatives has recently been presented. The novelty of the proposed method is that in the first phase a DES is formed from *o*-phenylenediamine (*o*-PDA) and choline chloride (ChCl) as components (Scheme 7) [75].

**Scheme 7.** Novel synthetic route for benzimidazoles involving DES (*o*-PDA: ChCl).

A differential scanning calorimetry (DSC) analysis of the obtained DES and of the individual components was performed. The analysis showed the formation of the DES. ChCl and *o*-phenylenediamine showed melting points at 302 °C and 102 °C, respectively, while the obtained mixture showed a eutectic that melts at 32 °C. This result (a melting point significantly lower than that of its single components) demonstrated the successful formation of a eutectic mixture (Table 4).

**Table 4.** Eutectic temperature ($T_f$) of the DES and melting point ($T_m$*) of the pure HBD.

| ChCl | HBD | Molar Ratio | $T_f$ (°C) | $T_m$ HBD (°C) | Δ (°C) | Appearance |
|---|---|---|---|---|---|---|
|  |  | 1:1 | 32 | 102 | 70 | Light yellow liquid that tends to become greenish. |

The obtained eutectic mixture was tested as solvent and, at the same time, reactant in the pilot reaction to obtain benzimidazole derivatives. To the DES synthetized, 1 mol of benzaldehyde (respect component *o*-PDA) was added and magnetically stirred for 10 min at 80 °C. The reaction obtained the compound mono substituted (95% yield) and compound disubstituted as the only product (97% yield) using 2 mol benzaldehyde.

Using DES solvent as starting material with different molar ratio of the aldehyde, various 2- substituted or 1,2-disubstituted benzimidazoles can be obtained in good selectivity and yields. An essential benefit of this solvent system is that the use of the DES enables an easy work-up without using any purification methods, thanks to the selectivity method.

## 3. Pharmacological Activity of Benzimidazoles

### 3.1. Antihypertensive Activity

In the research and development of antihypertensive medicines, the benzimidazole nucleus has been investigated thoroughly. Intercepting the renin–angiotensin system (RAS) is the mechanism that allows several benzimidazole drugs to function as antihypertensives [76,77]. The RAS cascade is responsible for the production of the active pressor known as angiotensin II (Ang II) [78,79]. Angiotensinogen, which is a polypeptide, is cleaved by renin to form a decapeptide known as Ang I. Ang I is then acted upon by angiotensin converting enzyme (ACE), which results in the production of Ang II.

Ang II interacts with angiotensin receptor 1 (AT1), causing vasoconstriction, Na+ retention, and aldosterone release, all of which contribute to hypertensive action. Preventing the synthesis of Ang II, as well as blocking the binding of Ang II to AT1 receptors, are two of the many methods that can be used to regulate the effects of angiotensin II. It has been shown that inhibition at the receptor level provides the highest levels of safety, specificity, and efficacy. Because of this, the production of antihypertensives have been concentrated on the development of AT1 receptor blockers [80,81], such as 2-butyl-benzimidazole-7-carboxylic acid derivative (Compound **1**, Figure 4) [82].

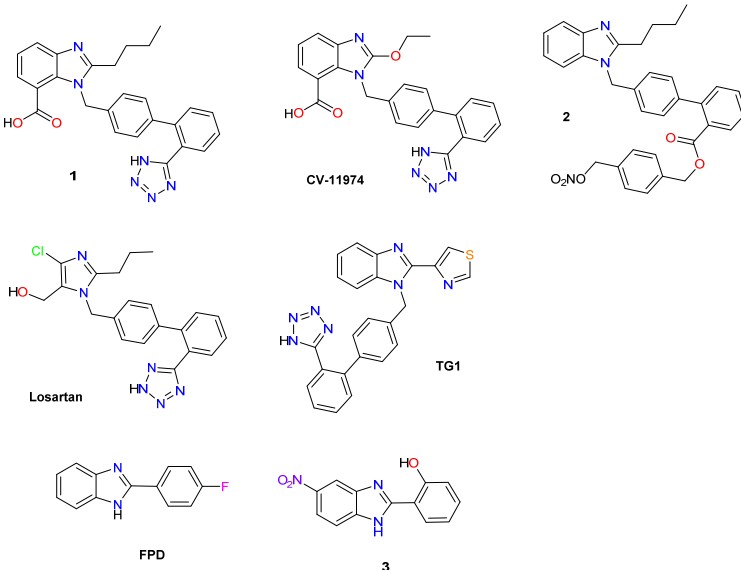

**Figure 4.** Selected examples of benzimidazole analogs bearing antihypertensive properties.

Optimization or substitution of the benzimidazole structure has resulted in the production of compounds with varying degrees of potency [83,84]. For example, **CV-11974** was produced by optimizing the functional groups of benzimidazole **1** that surround the heterocyclic backbone. This resulted in the production of a compound that lowers blood pressure in a dose-dependent manner by inhibiting AT1 receptors in a non-competitive manner due to its slow dissociation from AT1 receptors. It has a markedly higher activity level compared to **Losartan** [85,86]. It has also been discovered that position 4 must remain unsubstituted for there to be a good interaction between the N-3 of the nucleus and the H-bond donor site in the AT1 receptor. On the other hand, position 1 is conserved for the biphenyl moiety. Compounds with varying degrees of activity have been created by replacing the tetrazole moiety with a variety of acidic heterocycles [87,88]. Among them, tetrazole-containing derivatives are the only ones proved to be both effective and bioavailable in comparison to other compounds. The presence of a -COOH group at the 7-position was shown to increase further the antihypertensive activity [89].

A series of benzimidazoles were synthesized by Zhang and colleagues by coupling an NO-donor with a benzimidazole core. Among the synthesized compounds, derivative **2** exhibits the most impressive and longest-lasting Ang II antagonist ability [90].

In 2020, Zhu Wu and his colleagues came up with a design for a number of novel fluoro-substituted benzimidazole derivatives, all of which were capable of binding with the AT1 receptor, resulting in a considerable reduction in blood pressure. Compound **TG1** revealed a remarkable systolic blood pressure of 147.2 mm/Hg when tested for its antihypertensive efficiency [91].

Acute renal hypertension in guinea pigs was used to investigate a variety of 5-substituted benzimidazoles for antihypertensive efficacy in vivo. The series reported by Jain and colleagues [92] revealed extraordinary activity when compared to the conventional drug **Losartan**. Moreover, the activity was found to be rather better in the substituted alkylamino group at the 5-position of the benzimidazole backbone. As shown by Dorsch et al., a very effective benzimidazole-derived product may also be generated by the introduction of a pyridazinone moiety [93]. Using a variety of substituents such as amino, nitro, alkyl/arylsulfonamido, and alkylcarboxamido at position 5 can also considerably boost the activity of the target compounds [94,95].

Sharma et al. have successfully synthesized a large variety of sartans by inserting different substituents at the 2-, 5-, and 6-positions in tetrazolylbiphenyl or carboxylbiphenyl modified benzimidazole [96,97]. In order to treat hypertension in the SHR model, Iqbal and colleagues created the new angiotensin II receptor blocker fluorophenyl benzimidazole

(**FPD**) [98,99]. Estrada-Soto et al. have synthesized a series bearing a phenyl ring at the 2-position and varied substituents (–H, –CH$_3$, –NO$_2$, and –CF$_3$) at the 5- and 6-positions [100]. Derivative **3**, which has an IC50 value of 0.95 and a l M value of 2.01, has been shown to be the most effective compound among the former series [101].

*3.2. Anti-Inflammatory Activity*

The importance of the inflammatory component in numerous diseased states such as Crohn's disease [102], rheumatoid arthritis [103], asthma, multiple sclerosis [104], Alzheimer's disease [105], carcinoma, psoriasis [106], diabetes mellitus [107], viral infections, control of inflammation, etc. has become of prime importance in recent years [108,109]. Inflammatory mediators such as interleukins 1–16 (IL-1 to IL-16), tumor necrosis factor-a (TNF-a), prostaglandins, histamine, chemokines (CXC, CC, and C subsets), leukotrienes, serotonin, plasma proteases, and colony stimulating factors are some of the most common and widely explored points for controlling inflammation (CSF) [110,111]. These mediators are generated through a number of distinct processes that include cyclooxygenases, caspases, and kinases such as mitogen activated protein kinase 38 (MAP38) [112], cyclin dependent kinases (CDK1 and CDK5) [113,114], serine threonine kinases (IKK1 and IKK2), c-Jun N-terminal kinase (JNK) [115], and interleukin receptor associated kina (TNFK) [116,117]. Benzimidazole nucleus replaced at 1-position with various heterocycles has proved to yield strong anti-inflammatory products [118].

Compound **4** (Figure 5) of the series of 1-(substituted pyrimidin-2-yl)benzimidazoles has been shown to have an anti-inflammatory effect by blocking the activity of Lck [119]. Buckley et al. have brought another similar 1,6-disubstituted compound as a highly potent IRAK4 inhibitor with good TNFa inhibition [120]. Recent work by Makovec et al. reports the discovery of a variety of heterocyclic amidines, from which 2-phenyl-5-amidinobenzimidazoles may be produced and used to effectively suppress the generation of IL-1b [121]. Compound **5** featuring a pyridine ring replaced with an amino group demonstrated highly effective 5-lipoxygenase inhibition [122].

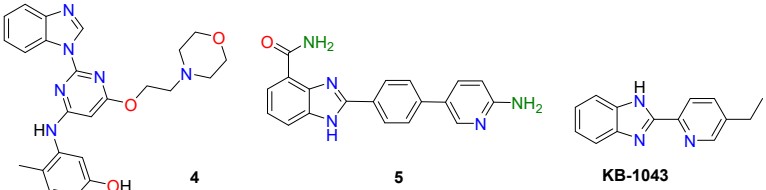

**Figure 5.** Selected examples of N-substituted or substituted at position 2 benzimidazole analogs bearing anti-inflammatory properties.

The Pharmaceutical Research Centre at Kanebo Ltd. In Japan has developed a series of 2-(2-pyridinyl)benzimidazoles by isosteric replacement of the thiazole ring. This was performed based on the moderate anti-inflammatory and analgesic activities of **Thiabendazole**, which is a well-known anthelmintic [123,124]. After testing 55 different compounds, 2-(5-ethyl-2-pyridinyl)benzimidazole (**KB-1043**) was found to possess anti-inflammatory, analgesic, and antipyretic properties that were superior to those of **Phenylbutazone** and **Tiaramide**. In addition to this, it causes less irritation to the gastrointestinal tract and has a therapeutic index that is two to three times higher than the reference compounds [125].

Taniguchi et al. have synthesized a variety of 2-aminobenzimidazole derivatives containing a wide variety of plausible substituents at the 1-position [126]. It has been reported that a structure-based design of 2-methyl-*N*-substituted benzimidazole containing a variety of sugar moieties (**6**) has strong anti-inflammatory action. This activity is dependent on the kind and linked-position of the sugar that is conjugated to the nucleus of the molecule [127].

SAR studies on approximately 50 compounds revealed that an electron rich group such as -methoxy, -ethoxy, -methylamino, or –dimethylamino on para position to the phenyl

ring increases the activity. Compounds that include a substituent in the 1-position have a greater capacity to reduce inflammation in the body.

Strong anti-inflammatory action was shown for compounds bearing electron-releasing methoxy group at position 6 and two-pyrrolidines (**7**, Figure 6). While compounds possessing electron-donating groups had lesser potency, those with electron-withdrawing nitro groups at position 6 were found to be more active [128].

**9** (R₁, R₂ = substituted or unsubstituted phenyl)

**Figure 6.** Selected examples of disubstituted and fused-ring benzimidazole analogs bearing anti-inflammatory properties.

Mader et al. [129] have created 2-amino-1-isopropylsulfonyl 6-substituted benzimidazole (**8**), which acts as a strong inhibitor of TNF-a and p38a MAP kinase. In comparison to the traditional medication **Diclofenac sodium**, they discovered that all of the synthetic compounds exhibited modest anti-inflammatory action. Strong anti-inflammatory action was produced by phenyls that were either left unsubstituted or substituted at (**9**) R₁ with 2-position electron-withdrawing groups or 4-position electron-donating groups. At R₂, however, an ortho phenolic substitution favored anti-inflammatory properties [130]. Among those, derivatives **10** and **11** were found to be the most promising anti-inflammatory candidates.

### 3.3. Antiviral Activity

Among the number of nucleoside benzimidazole analogues prepared through the years, 5,6-dichloro-l-(β-d-ribofuranosyl)benzimidazole (**DRB**) derivatives (Figure 6) [131,132] have gained momentum due to their activity as antiviral agents, in particular against human cytomegalovirus (HCMV) and other RNA viruses [133,134]. These benzimidazoles efficiently inhibit viral RNA synthesis by blocking RNA polymerase II. Notwithstanding their anti-viral activity, DRB derivatives are often characterized by high cytotoxicity, which precludes their use as drugs.

A plethora of non-nucleoside benzimidazole analogues have been prepared and reported to be active against different virus strains. Even though these benzimidazoles were inferior in activity to the benzimidazole ribonucleosides as agents against HCMV as a part of broader antiviral testing, these have been shown to be potent reverse transcriptase inhibitors (RTIs) against HIV-1 replication [135]. In this context, several amine substituted N-(1H-benzimidazol-2ylmethyl)-5,6,7,8-tetrahydro-8-quinolinamines were synthesized [136].

C2-substituted benzimidazole-N-carbamates were reported as potential antivirals against herpes simplex virus type 2 (HSV 2) and Coxsackievirus B2. The derivative **12** featuring isopropylcarboxamide group at 2-position was the most active at noncytotoxic concentrations [137]. The derivative **13** featuring carboxamidine substituent at C-5 and N-methyl-pyrol at C-2 showed prominent activity against all four tested types of viruses (Adenovirus 5, Herpesvirus 1, Coxsackievirus B5, Echovirus 7) while having no cytotoxic activity (Figure 7) [138].

The N-1-substituted benzotriazole **14** was identified as a representative of a class of respiratory syncytial virus (RSV) inhibitors. The structure-activity relationship examination established the essential nature of a side-chain moiety appended to the benzimidazole

nucleus for the expression of potent antiviral activity [139]. Further studies established that alternatively to the benzotriazole moiety the Benzimidazol-2-one derivatives **15** one forms a more effective structural background for the generation of potent inhibitors of RSV [140,141].

**Figure 7.** Selected examples of benzimidazole analogs with antiviral activity.

The 5,6-dichloro benzimidazole phthalimide derivatives based on an identified lead were prepared and assessed for their anti-hepatitis B virus (HBV) activity. In this series, most of the derivatives proved to be potential HBV inhibitors with high selectivity indices. Among them, compound **16** has been identified as the most promising [142].

Several other classes of benzimidazole derivatives, such as coumarin conjugates [143,144] and benzimidazole diamides [145,146], were considered as potential lead hepatitis C virus NS5B (HCV NS5B). Currently, **Pibrentasvir** is an inhibitor of the HCV NS5A protein approved in US and Europe.

Azzam and colleagues created a variety of novel substituted 2-pyrimidylbenzothiazoles and evaluated their antiviral potency. These compounds either featured sulfonamide moiety or an amino group at the C2 of the pyrimidine ring [147]. The novel ring structure was synthesized via Michael addition reactions of guanidine or *N*-arylsulfonated guanidine with different ylidene benzothiazole derivatives. Plaque reduction tests were utilized to evaluate the antiviral activity of the newly synthesized compounds against HSV-1, CBV4, HAV HM 175, HCVcc genotype 4, and HAdV7. With significant IC50, CC50, and SI values, it was shown that 5 out of the 21 synthesized compounds exhibited greater viral reduction in the range of 70–90% when compared to **Acyclovir**. In the instance of CBV4, nine compounds have shown a reduction of more than 50%. When evaluated against HAV, seven of these newly developed drugs had similar outcomes, with just a couple showing 50% or more decreases against HCVcc genotype 4. Surprisingly, one compound, 2-((5-(benzo[d]thiazol-2-yl)-2-((phenylsulfonyl)methyl)pyrimidin-4-yl)amino)-1-(4-bromophenyl)ethan-1-one (Figure 8, A), showed extensive activity against all five tested viruses, suggesting that it may be effective as an antiviral medication. The examined compounds interacted hydrophobically with the active site of Phe 138 through the pyrimidine ring of ACV (Figure 8). The molecule had a single hydrogen-bonding at a distance of 3.84 Å between the amino acid residue Asp 102 and the sulfur atom of the benzothiazole ring.

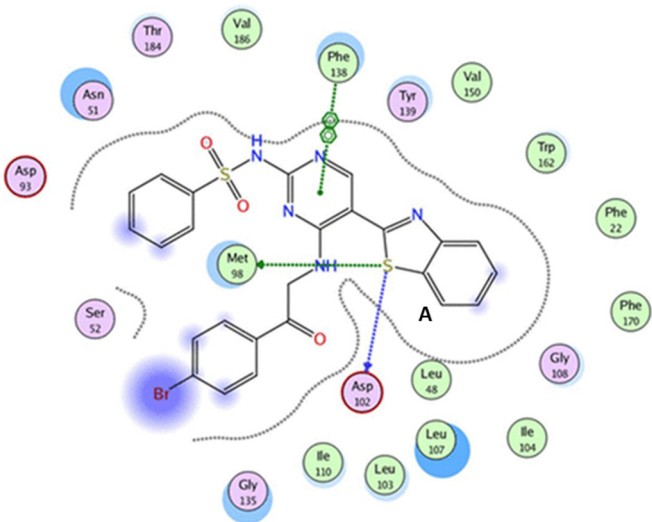

**Figure 8.** Best docked position of A within the binding pocket of Hsp90 (PDB ID 3b25).

*3.4. Antitumor Activity*

Many benzimidazole derivatives are used as antitumor agents. Tubulin polymerization inhibition properties of a range of benzimidazoles has been intensively studied [148,149]. In this series, BAL27862 (**Avanbulin**) is currently undergoing clinical trials as a water-soluble lysine prodrug of BAL101553 (**Lisavanbulin**) [150,151]. This drug is a potent inhibitor of tumor cell growth and shows both promising anti-proliferative and vascular-disrupting activity. **Nocodazole** is another notable example of a microtubule-targeting agent with antiproliferative activity that entered clinical trials (Figure 9) [152,153].

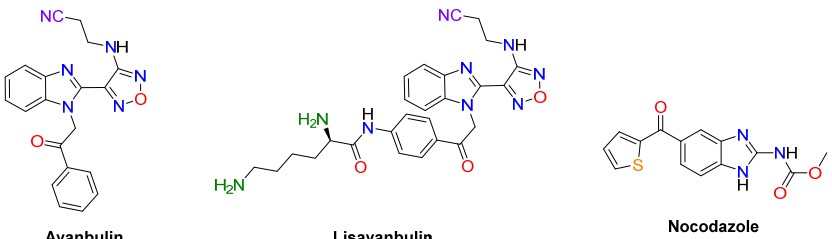

**Figure 9.** Selected examples of tubulin polymerization inhibitors.

The flexible nature of the bis-benzimidazole ring has been explored as a core in several DNA topoisomerase inhibitors. The benzimidazole **MH1** was reported as potent topoisomerase II inhibitor [154].

Many benzimidazole derivatives have been also shown to act as DNA minor groove binders [155]. Bis(benzimidazole) compounds are well-known to interact with DNA both by intercalation and by groove binding [156,157]. The extensive research in this area [158,159] led to the discovery of the dimeric bis-benzimidazole **17** with high binding affinity to the motif $[A,T]_4$-$[G,C]$-$[A,T]_4$ in the DNA sequence [160]. **Bendamustine** [161] is a notable example of an FDA approved DNA alkylating agent for the treatment of chronic lymphocytic leukemia and B-cell non-Hodgkin's lymphoma (Figure 10).

The kinase inhibitors are one of the most explored classes of benzimidazole chemotherapeutic agents [162,163]. The 4,5,6,7-halogenbenzimidazoles are potent protein kinase CK2 inhibitors. In this series, the iodine derivative **TIBI** (Figure 11) exhibits the best binding affinity [154–166]. **Abemaciclib** is currently FDA-approved for the treatment of advanced or metastatic breast cancers and operates by cyclin-dependent kinase (CDK) inhibition [167]. **Dovitanib** [168,169] is a multitargeted growth factor receptor kinase inhibitor also approved for the treatment of renal cell carcinoma (RCC) patients.

**Figure 10.** Selected examples of antitumor.

**Figure 11.** Selected examples of kinase inhibitors.

Benzimidazole derivatives have been reported as aromatase [170,171] and dihydro-folate reductase (DHFR) inhibitors [172,173]. **Liarozole** is a notable example of a potent aromatase inhibitor and retinoic acid metabolism-blocking agent that was shown to exhibit cytotoxicity against several types of cancer cells [174,175].

Wang and colleagues created and tested numerous 1-benzene acyl-2-(1-methylindol-3-yl)-benzimidazole derivatives aiming to investigate their cytotoxicity against human cancer cell lines and potential tubulin polymerization inhibitors [176]. Comparing the examined derivatives to the positive controls colchicine and CA-4, compound **11f** had the strongest tubulin polymerization inhibition activity (IC50 = 1.5 mM) and antiproliferative activity against A549, HepG2, and MCF-7 (GI50 = 2.4, 3.8, and 5.1 mM, respectively). Additional docking studies demonstrated the binding mode of compound **11f** interacting with 1SA0 protein. Docking results demonstrated that Ser178, Cys241, Leu248, and Lys352, located in the binding pocket of protein, played crucial roles in the conformation with compound **11f**, which was stabilized by two π-cation bonds and two hydrogen bonds as shown in the 2D diagram. In region A, an indole ring with five members and the amino acid Lys352 formed a 6.9-π-cation link. The second π-cation link with a 2.7 Å- distance was created by the amino acid Leu248 and the benzene ring C in compound **11f**. Ser178 of the amino acid formed a hydrogen bond with the methoxy oxygen atom of section A, whereas Cys241 of the amino acid formed a hydrogen bond with the oxygen atom of section C (Figure 12).

Hashem and colleagues employed microwave irradiation and 2-phenylacetyl isothiocyanate as the main starting material to design and synthesize a new series of benzimidazole, 1,2,4-triazole, and 1,3,5-triazine derivatives. Using **Doxorubicin** as the reference drug, all new analogues were evaluated as anticancer agents against several cancer cell lines. Most of the substances under investigation exhibited particular cytotoxic action against MCF-7 and A-549 cancer cell lines [177]. With an energy score of 10.88 kcal/mol, compound **6b** was determined to be the most successful inhibitor since it completed the crucial contacts in the EGFRWT active site. Here, a π-cation contact between the phenolic ring and the Val702 residue and a hydrogen bond between the side chain of Lys721 and the N-2 of the 1,2,4-triazole moiety were also established (distance: 3.63 Å). It is made easier for the enzyme to fit into its active site by the H-bond acceptor, which is situated between the

hydroxyl oxygen and the backbone of the significant amino acid Met769 (distance: 3.27 Å) (Figure 13).

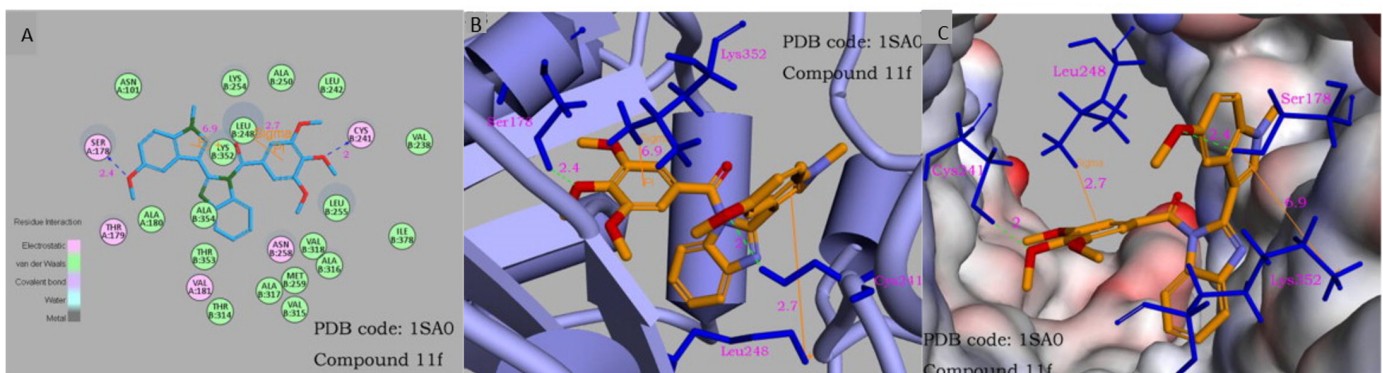

**Figure 12.** Interaction between tubulin and chemical 11f's active conformation. (**A**) A 2D schematic depicts the interaction of compound 11f with the colchicine binding site. The H-bond (blue arrows) is shown as dotted arrows, whereas the π-cation interaction is shown as orange lines. (**B**) A three-dimensional illustration depicts the interaction of compound 11f with the colchicine binding site. For convenience, only interacting residues are shown. Orange lines represent the p-cation interaction, while dot arrows represent the H-bond (green arrows) (**C**).

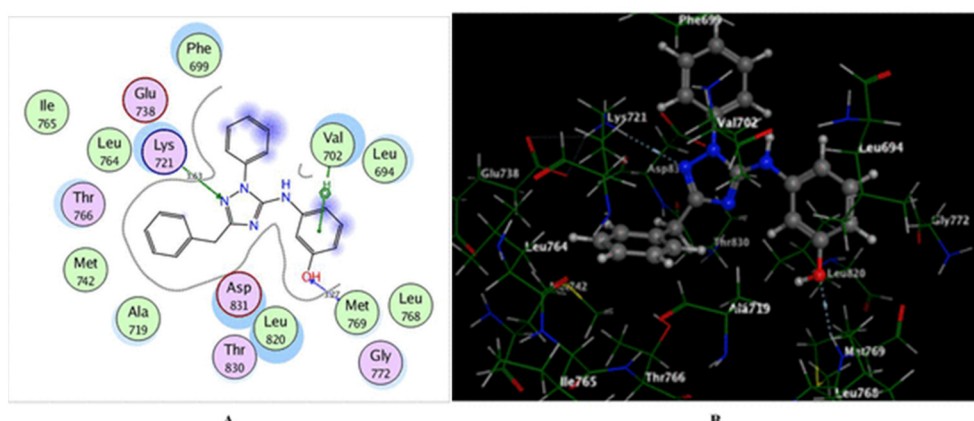

**Figure 13.** A 2D and 3D schematic of binding interactions (**A**,**B**) of 6b into EGFRWT.

A series of pyrazoles bearing a benzimidazole core were prepared using a multistep synthesis method starting form arylhydrazine and aralkyl ketones [178]. The hybrids were created, and the most potent one against the two pancreatic cancer cells (SW1990 and AsPCl) was found to be the one with a *para*-fluorophenyl unit tethered at the pyrazole nucleus (**6h**). The benzimidazole tethered pyrazole **6h** exerted its binding energy at a rate of 8.65 kcal/mol in addition to its predicted IC50 value of 457.41 nM. With the simple hydrophobic amino acid glycine (GLY 142) and the electrically charged acidic amino acids glutamic acid (GLU 149) and aspartic acid (ASP 108), the molecule **6h** shows van der Waals interactions. Tyrosine (TYR 105), leucine (LEU 134), and phenyl alanine (PHE 101, PHE 109, and PHE 150) are amino acids with hydrophobic groups. The hydrophobic and aliphatic amino acid alanine (ALA 146) and the secondary amino group of the imidazole scaffold were also shown to form a hydrogen bond. The aromatic ring tethered to the pyrazole structural motif interacts with the electrically charged and basic amino acid arginine (ARG 143) in this way. Additionally, a relationship between carbon and the charged amino acid, arginine (ARG 143), and the fluorine-containing aromatic moiety was observed. Additionally, there is a lone pair interaction between the aromatic nucleus that is directly coupled to the nitrogen of the pyrazole scaffold and glutamic acid (GLU 133),

an electrically charged acidic amino acid. In addition, the hydrophobic and aliphatic side chains of the amino acids methionine (MET 112) and valine (VAL 153) engage with the fused phenyl ring of the benzimidazole structural motif through alkyl interactions. The hydrophobic and aromatic amino acid leucine (LEU 134) and another aromatic nucleus tethered to the pyrazole scaffold, as well as the electrically charged and basic amino acid arginine (ARG 136) and the phenyl moiety integrated with nitrogen of the pyrazole unit, exhibit alkyl interactions (Figure 14).

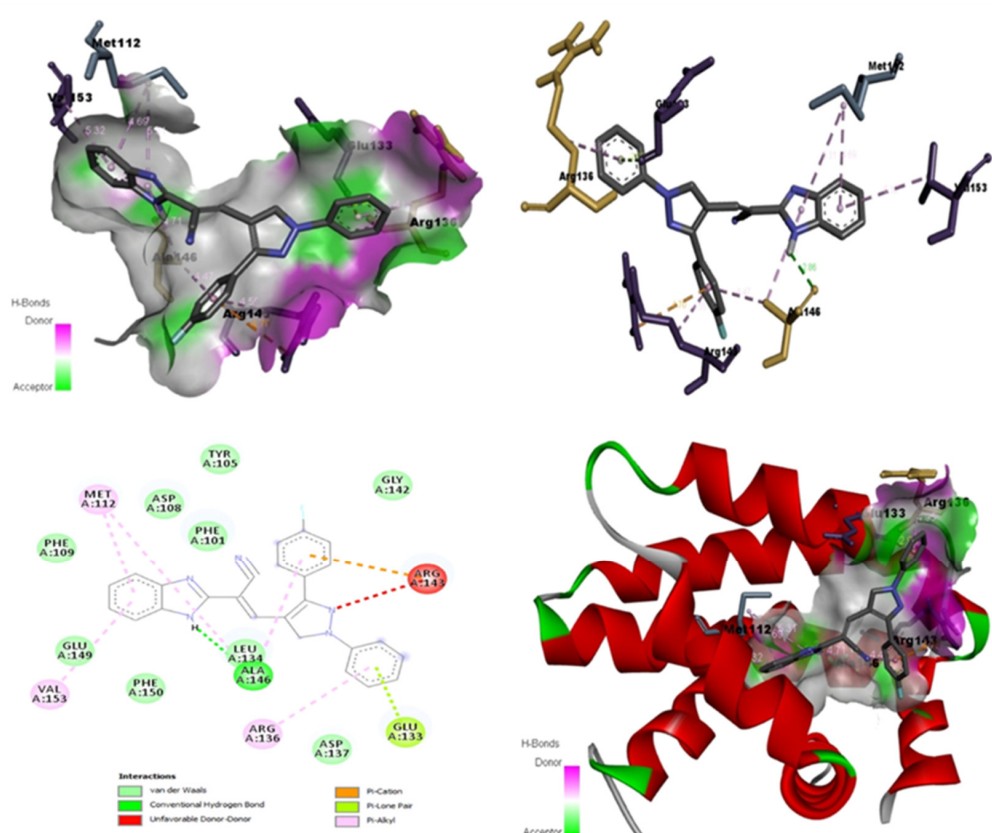

**Figure 14.** The 2D and 3D interactions of BCL-2 with compound **6h**.

According to El-Meguid and colleagues' research, 1-(6-benzoyl-2-(3,4-dimethoxypheny l)-1H benzo[d] imidazol-1-yl)propan-2-one was used to prepare a series of 6-benzoyl benzimidazole derivatives [179]. While the compounds were being investigated as cytotoxic agents against cervical cancer cells, **Doxorubicin** was used as a reference drug (Hela). The bulk of the compounds under investigation were both safe to utilize with normal cell lines and showed promising anticancer activities. The best options, as opposed to **Erlotinib**, were found to be EGFR, HER2, PDGFR-b, and VEGFR2 inhibitors. Two substances had promising suppressing effects, it appeared. Furthermore, it was shown that the latter compounds might cause cellular apoptosis, cell cycle arrest in the G2/M phase, and cell accumulation in the pre-G1 phase. The ligands fit into the HER2 active site with Glide docking scores consistent with their biological activity. The docking poses of the ligands had favorable effects on HER2's DFG region, catalytic loop, activation loop, G-rich loop, phosphate binding loop, and a-C helix. Among all examined compounds, derivative **13** had Glide scores of 9.7 kcal/mol and showed excellent binding site fitting. This compound appears to be actively interacting with Asn 850 of the catalytic loop and Asp 863 of the DFG region via hydrogen bonding. There are π-π interactions with Phe 864 of the DFG region (Figure 15).

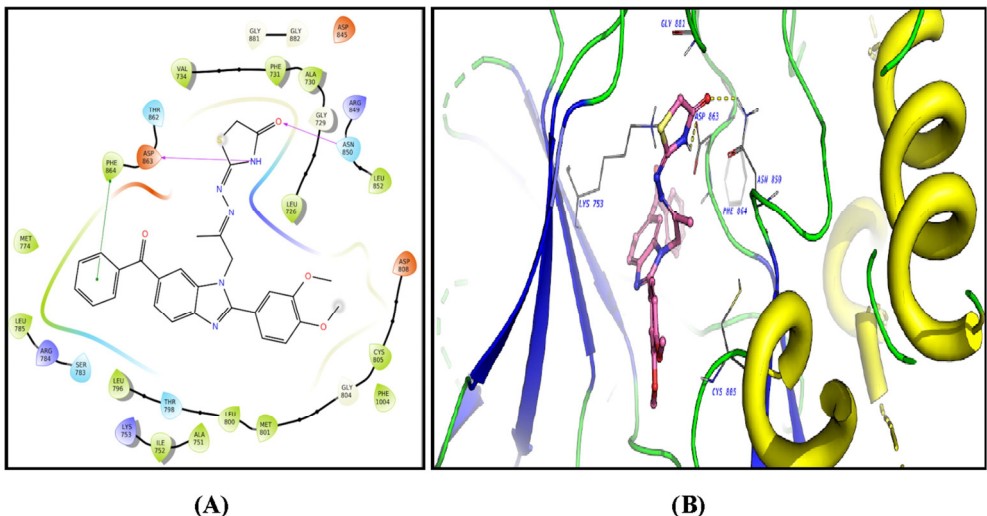

**Figure 15.** Compound **13** is shown in a 2D figure (**A**) and a 3D depiction (**B**) that demonstrate how it interacts with the HER2 kinase active site.

### 3.5. Antioxidant Activity

The homeostatic balance between reactive oxygen species (ROS) and endogenous antioxidants plays an important role in maintaining healthy tissues. Under oxidative stress, human bodies produce excessive ROS, damaging various cell constituents (such as DNA, lipids, and proteins), consequently causing cell death and health problems.

Increasing evidence shows that ROS-induced oxidative damage facilitates the development of a variety of diseases including myocardial infarction, cancer, neurodegenerative disorders, and inflammation [180]. Therefore, antioxidants are very important for protecting living organisms against excessive ROS, and there is special scientific interest to develop diverse types of antioxidants for medical treatment.

In the last decade, the antioxidative potency of numerous benzimidazole derivatives has been studied, focusing on incorporating different substituents on the phenyl ring or heterocycles into the benzimidazole skeleton and on forming complexes with metals.

### 3.5.1. Incorporation of Heterocycles

Some heterocycles incorporated into the benzimidazole skeleton to improve its antioxidant properties are pyrrole, pyrazole, indole, thiophene, etc.

Kosolapov et al. (2013) studied the antioxidant activity of pyrrolo [1,2-α] benzimidazole derivatives on four model free-radical systems in comparison to the reference antioxidant Trolox. Compound **18** showed excellent antioxidant activity (Figure 11) [181]. Similar results were obtained by Djuidje and colleagues in 2020. Specific features of the chemical structure of pyrrolo [1,2-α] benzimidazole derivatives, significant π-redundancy, conjugated electron density of the molecule, and the presence of pyrrole ring attest to the high antioxidant potential of these substances [182].

On the other hand, a series of new N-substituted pyrazole-containing benzimidazoles were synthesized by Bellam [183]. Some of these derivatives with benzyl substituent at the imidazole nitrogen exhibited good antioxidant activity. Compounds containing the benzyl group as a substituent on imidazole nitrogen, especially compound **19**, exhibit good activity, which may be attributed to the extended resonance by the radical that is formed from the benzyl group upon reaction (Figure 16).

Mentese et al., in 2013 [184], synthesized a novel 1H-benzo[d]imidazole derivative containing 1,2,4-triazole (**20,** Figure 16), which showed anti-lipase activity in addition to a good performance on free radical scavenging assays.

In 2014, Ujjwal et al. reported the synthesis of a series of 1,3,4-oxadiazole derivatives bearing benzimidazole nucleus (**21**, Figure 16) that were found to be moderately active

at higher concentrations as compared to ascorbic acid in 2,2-diphenyl-1-picryl hydrazide (DPPH) method [185].

**Figure 16.** Pyrrolo, pyrazole, triazole, and oxadiazole–benzimidazole derivatives.

Similarly, ALP et al. (2015) [186] revealed a selective synthesis of 2-(substitutedbenzylth io)-5-((2-(4-substituted phenyl)-1H -benzo[d]imidazol-1-yl)methyl)-1,3,4-oxadiazoles deriva- tives (**22**, Figure 16). The compound, which bears benzyl groups, showed the best antioxi- dant activity.

Similarly, a series of 2-(Thiophen-2-yl)-1-((thiophen-2-yl)methyl)-1H-1,3-benzimidazole (**23**) were synthesized by Latha et al. (Figure 17), and antioxidant property and DNA bind- ing behavior of the compound has been investigated, exhibiting moderate antioxidant activity [187].

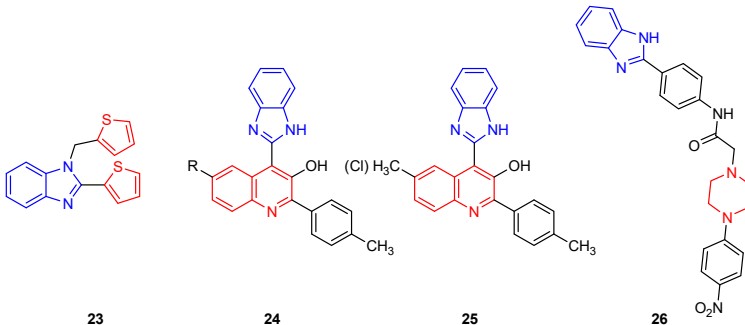

**Figure 17.** Thiophenyl, quinolin-2-ols, benzohydrazone, thiadiazol-5-yl, piperazine–benzimidazole derivative structures.

In 2020, Massoud et al. reported the synthesis of 4-(1H-benz[*d*]imidazol-1-yl)-3-phenyl- 6-substituted quinolin-2-ols (**24**, Figure 17), exhibiting a slightly higher antioxidant activity than ascorbic acid [188].

Studies accomplished by Ramprasad et al. (2015) demonstrated that compounds derived from 2-(imidazo [2,1-b][1,3,4]thiadiazol-5-yl)-1H-benzimidazole with chlorine and methyl as substituents (**25**) have an antioxidant capacity comparable with that of the reference standard BHT (Figure 17) [189].

Novel benzimidazole derivatives have been synthesized by Abbhi et al. (2017) through the reaction of 2-aminobenzimidazole with substituted piperazines using acetamide as the linker. The nitrophenyl piperazine substituted compound (**26**, Figure 12) showed a considerable free radical scavenging activity in the DPPH assay [190].

Spasov et al. (2017) synthesized novel 9-N-substituted derivatives of 2-(4-biphenyl-4-yl)-imidazo [1,2-*a*]-benzimidazole (**27**, Figure 18). The new compounds were found to exhibit multitarget action towards a number of targets and are promising for extended pharmacological studies of the antioxidant, PTP1B inhibitory, AMPK activating, and antiplatelet activities [191].

**Figure 18.** Imidazo–benzimidazole derivative structures.

In 2018, Ashok et al. synthesized a new series of triheterocycles containing indole-benzimidazole-based 1,2,3-triazole hybrids, with antioxidant activity (**28**, Figure 18) [192].

Similarly, Gullapelli et al., in 2021, synthesized a novel 2-(1-((1-substitutedphenyl-1H-1,2,3-triazol-4-yl)methoxy)ethyl)-1-((1-substitutedphenyl-1H-1,2,3-triazol-4-yl)methyl)-1H-benzo[d]imidazole derivative with important antioxidant activity [193].

In Alzheimer's disease, many chemical changes take place in the brain. One of the earliest and biggest changes is that there is less of a chemical messenger called acetylcholine (ACh). ACh helps the brain to work properly. Tacrine (TAC) slows the breakdown of ACh, so it can build up and have a greater effect.

A series of tacrine-hydroxyphenyl benzimidazole (TAC-BIM) hybrid compounds (**29**, Figure 18) were developed and assessed. Furthermore, these compounds exhibited moderate antioxidant (radical scavenging) capacity but the best activity was found in the compound which includes a hydroxyl group in the linker.

Therefore, these hybridization strategies revealed success and some compounds are worthy of further development as potential multifunctional drug candidates for Alzheimer's disease (AD) therapy [194].

On the other hand, a series of new 2-(4-nitrobenzyl)-1H-benzimidazole derivatives containing 1-substituted thiosemicarbazide, triazole, oxadiazole, thiadiazole, and imine groups were synthesized and showed significant antioxidant activity. Additionally, among the tested compounds, (5-{[2-(4-Nitrobenzyl)-1H-benzimidazol-1-yl]methyl}-N-phenyl-1,3,4-thiadiazol-2-amine, **30**) displays the best inhibitory effect against xanthine oxidase (Figure 18) [195].

Similarly, Guner et al. (2019) synthesized novel 2-substituted benzimidazole molecules with triazole, thiadiazole, and oxadiazole rings showing an increase in antioxidant levels [196].

A series of functionalized benzimidazole derivatives bearing N-(4-halophenyl)pyrrolid in-2-one moiety were synthesized by Tumosienė et al. N'-[(2-chloro-5-nitrophenyl)methylid

ene]-2-[2-[1-(4-chlorophenyl)-5-oxo-3-pyrrolidinyl]-1H-benzimidazol-1-yl]acetohydrazide (**31a**, Figure 19) exhibit significant antioxidant activity, higher than that of ascorbic acid [197].

**Figure 19.** Triazole, thiadiazole, and oxadiazole–benzimidazole derivative structures.

A series of benzimidazole compounds containing piperazine or morpholine cycle was designed and efficiently synthesized (**31b**, Figure 19). The results showed that most of the compounds demonstrated very high scavenging activity using CUPRAC, FRAP, DPPH, and ABTS methods. It is clear that the benzimidazole-piperazine (or morpholine) combination has the potential to serve as a pharmaceutical source for therapy [198].

Various benzimidazole entities linked to pyrazolyl and hydrazonoyl cyanide substrates carrying aryl and heteroaryl groups (**32**, Figure 19) were synthetized by Khalifa and colleagues in 2019. These novel compounds presented good antioxidant and scavenging activities, which could be explained because resonance phenomena of double bonds and lone pair electrons on nitrogen may lead to radical formation in more than one site, especially on the benzene ring attached to the nitro group (highly electron-withdrawing) which enabled the benzene ring to convert to a radical form and form a new covalent bond with another radical [199].

Abdelgawad et al. synthesized novel hybrid structures of benzimidazole with a pyrimidine scaffold (**33,** Figure 19). These new compounds had been evaluated for their antioxidant and anticancer activities exhibiting good scavenger effects [200].

Recently Shatokhin et al. synthesized hybrid molecules comprising chromone and benzimidazole rings and sterically hindered phenol fragments. Compound **34** exhibits antioxidant activity comparable to the reference drug Trolox (Figure 19) [201].

Arya and colleagues synthesized different coumarin-benzimidazole hybrids, studying the effect of different linkers between heterocycles. Compounds bearing thiometylene (**35**), amide (**36**), methylene-amide (**37**), and the methylene (**38**) linkers were the most active, displaying substantial free radical scavenging activity, comparable to that of the standard compound butylated hydroxytoluene (BHT) (Figure 20) [202].

**Figure 20.** Coumarinebenzimidazole and thiazino [3,2-a] benzimidazol-4-one hybrids structures.

Thiazines are very useful compounds in medicinal chemistry and have been reported to exhibit a wide variety of biological activities such as antimicrobials, antivirals, anti-

fungals, antihistamines, and antioxidants. Ramos Rodríguez et al. (2020) synthesized 2-aryl-2,3-dihydro-4H-[1,3]thiazino [3,2-a]benzimidazol-4-one with high anti-radical activity (**39**, Figure 20) [203].

### 3.5.2. Transition Metal Complexes

On the other hand, the transition metal complexes with benzimidazole have many potential applications. Consequently, Fu et al. (2014) studied a ternary copper(II) complex derived from 2-(20-pyridyl)benzimidazole) and glycylglycine, (**40**, Figure 21). It was found that the complex partially intercalated to DNA with high affinity. The complex cleaved DNA efficiently at a very low concentration ($\sim$5 $\mu$M) via an oxidative mechanism in the presence of Vc, with the HO. and O2$^-$. as the active species, and the SOD as a promoter [204].

**Figure 21.** Transition metal complexes with benzimidazole.

Islas et al. (2014) reported the synthesis of a new Cu(II) complex with the antihypertensive drug telmisartán (Tlm) (**41**, Figure 21), $[Cu_8Tlm_{16}]\cdot24H_2O$ (CuTlm). The antioxidant measurements indicate that the complex behaves as a superoxide dismutase mimic with improved superoxide scavenger power, as compared with native sartan [205].

Similarly, Wu et al. (2014) studied other copper(s) complexes with the benzimidazole ligands. They reported three copper(II) ternary complexes with the V-shaped ligand bis(2-benzimidazolylmethyl)amine (bba), with composition $[Cu(bba)(crotonate)].ClO_4$, $[Cu(bba)(methacrylate)].ClO_4$ and $[Cu(bba)(acrylate)(CH_3OH)].NO_3.H_2O$. Moreover, the antioxidant assay in vitro also shows that the copper(II) complexes possess significant antioxidant activities, with complex **42** (Figure 21) being the most active of the series [206].

Likewise, a series of ligand complexes containing M = Ni(II)/Cu(II)/Zn(II) ions with the ligands 5-Fluorouracil(5-FU: A) benzimidazole were synthesized and the antioxidant activity was evaluated, **43** (Figure 21). The results show that the mixed ligand complexes have more potent activities than free ligands [207].

Benhassine et al. synthesized similar complexes. The complexes containing Zn in their structure presented higher antioxidant activity [208].

Similar transition metal complexes were studied by Wu et al. (2015). Bis(N-ethylbenzimidazol-2-ylmethyl)aniline with Cu (**44**, Figure 17), Mn, and Ni exhibit good hydroxyl radical scavenging activity [209].

Both 2-aminobenzimidazole with 2,4-dihydroybezaldehyde (H3L) (**45**, Figure 22) and its Cu(II), Ni(II), and Co(II) complexes have been studied. The antioxidant activity decreases in the order Co(II) > free ligand >Cu(II) > Ni(II) [210].

Numerous studies have demonstrated that ligand L (**45**, Figure 22) and Ag(I) complex both bind to DNA via an intercalative binding mode, and the affinity for DNA is stronger in the case of Ag(I) complex when compared with ligand L, in a similar manner to that which occurs with the Cu Ln complex. Moreover, Ag(I) complex also exhibited potential antioxidant properties in vitro studies [211–213].

A series of C,N-cyclometalated osmium arene complexes containing benzimidazoles as ligands has been studied (**46**, Figure 22). Biological studies showed that complexes exhibited antioxidative properties by decreasing the levels of intracellular ROS and reducing the NAD+ coenzyme [214].

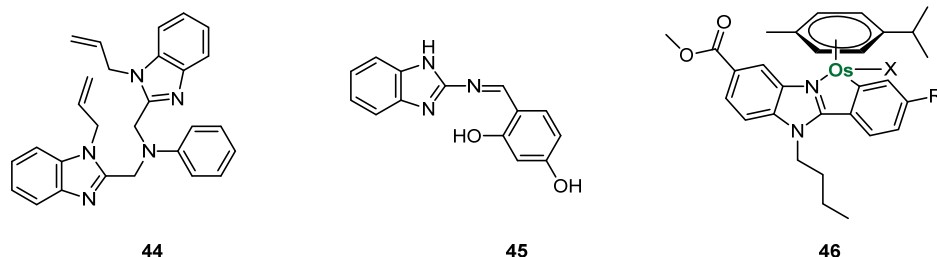

**Figure 22.** Transition metal complexes with benzimidazole.

3.5.3. Phenyl Ring Substitutions

The main modifications are related to the incorporation of hydroxyl groups in the benzimidazole structure.

In 2013, Khubaeva and colleagues synthesized a series of 1-(3,5-di-*tert*-butyl-4-hydroxy phenyl)- 2 arylbenzimidazoles. Benzimidazole derivatives with sterically hindered phenol substituent containing hydroxyl group in the *ortho* position with respect to the hindered phenol fragment presented the highest antioxidant activity [215].

In the same year, Rusina and Zhou synthesized a series of derivatives of 5_hydroxybenzimidazole (HBI) enter as a fragment into the composition of vitamin B12. They observed that the 5-hydroxy group is crucial for the antiradical activity of HBI [216,217].

The combination of hydroxyl groups with fluorine in the structure of benzimidazole seems to enhance antioxidant activity. This was demonstrated by the studies carried out by Shintre. Methyl 1-(4-fluorophenyl)-2-(2,3,4-trihydroxyphenyl)-1*H*-benzo[d]imidazole-5-carboxylate and Methyl 1-(4-fluorophenyl)-2-(2-hydroxy-4,6-dimethoxy-phenyl)-1*H*-benzo [d]imidazole-5-carboxylate showed good antioxidant activities, comparable to that of ascorbic acid [218].

Cindrić et al. synthesized novel methoxy amidino benzimidazoles derivatives. The compound bearing three methoxy groups on a phenyl ring and imidazolinyl amidine group on benzimidazole nuclei (5(6)-(2-Imidazolinyl)-2-(3,4,5-trimethoxyphenyl) benzimidazole hydrochloride) directly attached to the phenyl ring exhibited the most prominent reducing activity as well as free-radical scavenging activity [219].

In 2019, Anastassova et al. studied a series of N, N′-disubstituted benzimidazoles containing methoxy and hydroxyl substituents. The structural modification led to improvements in in vitro antioxidant activity. All of the compounds showed ABTS scavenging activity, while in the DPPH assay only the hydroxyl compounds were effective [220].

Three series of arylbenzimidazole derivatives have been simply synthesized and tested for their antioxidant capacity, by Baldisserotto. The 2-arylbenzimidazoles were tested against various radicals by the DPPH, FRAP, and ORAC tests and showed different activity profiles. It has been observed that the number and position of the hydroxy groups on the 2-aryl portion and the presence of a diethylamino group or a 2-styryl group are related to a good antioxidant capacity [221].

On the other hand, Sharma et al. synthesized a series of 5-methanesulfonamide benzimidazole derivatives. Compounds containing butyl, pentyl, and hexyl groups exhibited good antioxidant effects [222].

*3.6. Anticoagulant Activity*

Cardiovascular diseases (CVDs) are the leading cause of death worldwide. Therefore, scientists have exerted considerable effort in researching different kinds of drugs to treat CVDs.

Thrombin is a multifunctional specific trypsin-like serine proteinase that is crucial in the coagulation cascade. Therefore, thrombin is an attractive target for CVD therapy. Antithrombotic drugs have been recognized as the best for the treatment of CVDs and their complications.

The therapeutic landscape for anticoagulation management underwent a change from the use of traditional anticlotting agents such as heparins and warfarin [221] as the only options to the growing adoption of newer target-specific oral anticoagulants (NOACs) with

novel mechanisms of action [223–230]. Dabigatran (**47**, Figure 18), the first NOAC approved for use in the United States, is a direct competitive inhibitor of thrombin. Owing to benefits in ease of administration, safety, and efficacy demonstrated in clinical trials, Dabigatran was approved by the Food and Drug Administration (FDA) in 2010 via the accelerated pathway after a 6-month review. The haste to approve novel drugs places increasing importance on post approval data to help better understand risks and benefits [231].

Calkins et al. evaluated the efficacy of Dabigatran Etexilate compared to Warfarin in pulmonary vein ablation. They concluded that dabigatran was associated with significantly fewer bleeding complications than warfarin [232].

On the contrary, Rockey and Hernandez found that dabigatran seems to substantially increase the risk of gastrointestinal bleeding compared with warfarin in patients with atrial fibrillation (AF) [233,234] and increased bleeding and stroke compared with warfarin in patients with mechanical heart valves [235].

Böhm and Abraham analyzed changes in renal function in patients with AF who were assigned to receive either dabigatran or warfarin in the RE-LY trial (randomized evaluation of long-term anticoagulation therapy) [236,237].

These findings led researchers around the world to design synthetic modifications to the dabigatran molecule to reduce its unwanted effects.

So, the chemical class of benzimidazole derivatives is promising for the identification and development of effective platelet aggregation inhibitors.

In order to identify the structural features required for anticoagulant efficacy, Li et al. studied the structures of existing drugs by docking their molecular structures with the protein receptor. The structural features include: (1) there is an open-chain carboxylic acid attached to amine group; (2) the sites for hydrophobic interactions, electrostatic interactions, and hydrogen bonding interactions; (3) the presence of a carbonyl group and a terminal amidine which can form hydrogen bonding interactions and electrostatic interactions with amino acid residues [238].

A number of dabigatran derivatives were designed, synthesized, and evaluated as thrombin inhibitors.

When the different alkyl side chains were substituted at N-1 of benzimidazole, they influenced the inhibitory activities of the resultant molecule. This may be because of the limited steric effects. This trend demonstrates decreases in activity with increasing length or size of side chains.

The methyl and ethyl group can improve all the active pockets and have hydrophobic/lipophilic interactions with the surrounding amino acid residues. It was found ethyl group substitution at N-1 of benzimidazole exhibited better anticoagulant activity against thrombin (**48 a-b**, Figure 18) [239].

Complementary studies carried out by Yang et al. [240,241] established that introducing two fluorine atoms at the phenyl ring improved anticoagulant activity, as well as introducing fluorine at the para position, whereas the methyl group at this position inhibits this activity (**49**, Figure 23).

Bharadwaj et al. studied the beneficial role of benzimidazole-containing quinolinyl oxadiazoles on thrombotic disorders. The compound 2-(2-(5-Bromothiophen-2-yl)-1-methyl-1H-benzo[d]-imidazol-5-yl)-5-(2-(3,5-difluorophenyl)quinolin-4-yl)-1,3,4-oxadiazole exhibited anticoagulant and antiplatelet properties [242] (**50**, Figure 24).

In patients with peptic ulcer bleeding, an intravenous (IV) proton pump inhibitor (PPI) has been primarily used for the prevention of recurrent or delayed bleeding.

Omeprazole [243] was the first registered PPI and is one of the most frequently prescribed drugs worldwide; thus, its interaction with clopidogrel has been the most extensively characterized.

**Figure 23.** Dabigatran and its analog structures.

**Figure 24.** Quinolinyl oxadiazole benzimidazole derivative.

Subsequently, other PPIs were developed and launched on the market, including lansoprazole, pantoprazole, etc. [244]. The next advancement was the introduction of a new generation of long-term-acting PPIs, characterized by extended plasma half-life.

Spasov et al. studied antithrombotic activity of compound DAB-15 and compared it to known antiplatelet agents (acetylsalicylic acid, ticlopidine, and clopidogrel) in rat experimental model of arterial thrombosis induced by ferric chloride. The results suggest that compound DAB-15 exerted a dose-dependent antithrombotic effect and was superior to acetylsalicylic acid, ticlopidine, and clopidogrel [245].

During the past decades, clarifications of the mechanisms of heparin and progress in oligosaccharide chemistry have facilitated the development of synthetic oligosaccharides [246]. Synthetic oligosaccharides also exhibit antiplatelet properties. Encouraged by these observations to discover new biologically active benzimidazole oligosaccharide compounds, Yen et al. developed a new derivative, M3BIM, from D-maltotriose to achieve stronger antiplatelet effects. This study provided the first evidence that the novel compound M3BIM can be developed further into a new class of antiplatelet agents [247] (**51**, Figure 25).

**Figure 25.** Oligosaccharide benzimidazole derivative.

*3.7. Psychoactive Agents*

The number of new psychoactive substances (NPS) that have emerged on the European market has been rapidly growing in the last years, with a particularly high number of new compounds from the group of synthetic cannabinoid receptor agonists. There have been various political efforts to control trade and the use of NPS worldwide.

In this new act two groups of substances were defined: the group 'cannabimimetic/synthetic cannabinoids', covering indole, indazole, and benzimidazole core structures, and a second group named 'compounds derived from 2-phenethylamine'.

Before 2019, the solely available benzimidazole opioid was etonitazene (**52**, Figure 26), an extremely potent opioid, originally developed as a veterinary anesthesia, but never approved for medical use. Isotonitazene (**53**, Figure 26), has about half the potency of etonitazene, and several deaths involving isotonitazene (N,N-diethyl-2-[5-nitro-2-({4-[(propan-2-yl)oxy]phenyl}methyl)-1H-benimidazol-1-yl]ethan-1-amine) have occurred since it started circulating in the United States and Europe [248,249]. The substance was quickly risk-assessed by the EMCDDA, and the European Commission issued legislation banning isotonitazene in all member states [250].

**Figure 26.** Etonitazene, isotonitazene, and etonitazepyne structures.

In 2020 Blanckaert and colleagues reported on the identification and full chemical characterization of isotonitazene (**53**, Figure 26), a potent NPS opioid and the first member of the benzimidazole class of compounds to be available on online markets [251].

Based on this work, they managed to synthesize another analog, "etonitazepyne" (2-(4-ethoxybenzyl)-5-nitro-1-(2-(pyrrolidin-1-yl) ethyl)-1H-benzo[d]imidazole) (**54**, Figure 26), a novel member of the benzimidazole class of NPS opioids that structurally arises from cyclization of the two ethyl substituents of etonitazene to a pyrrolidine [252].

## 4. Conclusions

In this manuscript, a collection of eco-sustainable syntheses of benzimidazoles useful for the industrial production of drugs has been reported. In this regard, the known drugs derived from benzimidazole, their structure, and their pharmacological properties have been reported.

**Author Contributions:** M.N., N.C.H.C. and S.S. designed the original idea, wrote, and reviewed the manuscript. R.B., A.K., R.S., D.W. and A.P. wrote and reviewed the manuscript. All authors have read and agreed to the published version of the manuscript.

**Funding:** This research received no external funding.

**Data Availability Statement:** Data sharing is not applicable to this article.

**Acknowledgments:** We gratefully acknowledge financial support from the National Scientific Program "VIHREN" (grant КП-06-ДВ-1); Consejo Nacional de Investigaciones Científcas y Técnicas (CONICET) and Secretaría de Ciencia y Tecnología (SECYT-UNC).

**Conflicts of Interest:** The authors declare no conflict of interest.

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
