# Peer review of "A Review on the Green Synthesis of Benzimidazole Derivatives and Their Pharmacological Activities"

_catalysts, doi:10.3390/catal13020392_

Round 1
Reviewer 1 Report
Reviewer comments for catalysts-2146403
After careful review of this work, I found that review manuscript entitles “A review on the green synthesis of benzimidazole derivatives and their pharmacological actions.” by the authors, presented a good collection of work but it is missing many newer concepts for said derivatives.
The authors should revise their whole manuscript with the below highlight points:
Many numerous typographical errors within the manuscripts were found.
--In introduction part, author should mention the diverse synthetic protocols and used different effective catalysts for the said derivatives.
-- In my opinion, Figure 1 should be replaced with covered most potent benzimidazole derivatives with different targets ( at least 5 scaffoolds).
---We have found many missing latest references based green synthetic protocols for benzimidazole derivatives and their diverse pharmacological properties. Therefore, I advise to see the references based on synthesis of novel benzimidazole derivatives using new catalyst and should be discuss in introductory part e.g. Org. Med. Chem. Lett. 2014, 4 (1), 14 and ACS Omega, 2022,7(42), 36945-36987 and other missing references.
--The authors should go for careful proofread so as to eliminate (a) many typos; (b) grammatical errors; and also (c) remove unnecessary information and description.
In my opinion, this manuscript should not be accepted in its current form. It should be revised and must be added fresh collections of article with covering past 2- 5 years articles.
Author Response
According to the opinion of Reviewer 4, Figure 1 should be omitted form the text. Thus, we removed it and renumbered all figures through the manuscript accordingly.
Reviewer 2 Report
This review article concerns the green synthesis and biological activity of 2-aryl-substituted benzimidazoles. Although the chemistry of such scaffolds has received extensive studies and plenty of review work has been previously reported but there is always new. The work possesses some interest but it needs comprehensive revision before being recommended for publication, thus:
1- Authors foxed on synthesizing such attractive scaffolds via the reaction of aromatic aldehydes with o-phenylenediamine in different green reaction conditions, however, there are diverse other green efficient protocols ( e.g. Kumer et. al. Green Chem. ,21,3666 (2019); Kapoor et. al., Synlett, 18, 2809, Su et. al. J. Chem. Res. 333 (2009) ............... Authors should survey the other green protocols for green synthesizing such scaffolds.
2- I messed in the biological section the docking studies and mechanism of action at least for at least some biologically relevant products.
Author Response
We would like to express our gratitude to the Reviewer for the valuable suggestions in attempt to improve our manuscript. Below are listed the answers to the questions and remarks.
- I messed in the biological section the docking studies and mechanism of action at least for at least some biologically relevant product
- Thank you for your suggestion. In current revised version of our manuscript, we included several examples of docking studies and potential mechanisms of action. Please see 12,13,14 and 15.
Reviewer 3 Report
“A review on the green synthesis of benzimidazole derivatives and their pharmacological actions “ by Monica Nardi et al. contains complete and detailed information on the synthesis, structure and properties of a large number of benzimidazole derivatives. These data are of particular interest because benzimidazole derivatives are very widely used as terapeutic agents. Many of these unique compounds are used as antihypertensive, anti-inflammatories, antiviral, analgesics, anticancer, antifungals, antiparasitics, and antiulcers drugs. The review describes methods for the synthesis of these derivatives using of catalysis and green solvents. A number of imidazole derivatives with antihypertensive, anti-inflammatory, antiviral, antitumor, antioxidant, anticoagulant activity are described in detailed. The list of references contains 241 items, including the research of year 2022. The review is well written. It may contribute to the further development of work in this important direction and may be published in its present form after minor corrections (see remarks).
Remarks.
1. Page 8. line255: Рrinted: Intercepting the Renin–Angiotensin System is the mechanism that allows several benzimidazole drugs to function as antihypertensives (RAS) [72,73].
Should be: Intercepting the Renin–Angiotensin System (RAS) is the mechanism that allows several benzimidazole drugs to function as antihypertensives [72,73].
2. Page 9, line 292: There is no reference after “The series reported by Jain and colleagues”.
3. Page 12, line 394: “The 12gainst12aze 14” ? To explain.
4. Page 16, Figure 12. Compounds 24 and 25: The structures are the same.
5. Page 18, Figure 15 and line583. Compounds 37 and 38. The linker titles are messed up.
6. Page 19, line 626 and Figure 17: compound 44: to check the title.
Author Response
We would like to express our gratitude to the Reviewer for the valuable suggestions in attempt to improve our manuscript. Below are listed the answers to the questions and remarks.
- Page 8. line255: Рrinted: Intercepting the Renin–Angiotensin System is the mechanism that allows several benzimidazole drugs to function as antihypertensives (RAS) [72,73]. Should be: Intercepting the Renin–Angiotensin System (RAS) is the mechanism that allows several benzimidazole drugs to function as antihypertensives [72,73].
- Appropriate correction was applied to the manuscript text as per Reviewer’s suggestion.
- Page 9, line 292: There is no reference after “The series reported by Jain and colleagues”.
We apologize for the mistake. The reference was added to the text. https://link.springer.com/article/10.1007/s00044-012-0462-7
- Page 12, line 394: “The 12gainst12aze 14” ?To explain.
- We apologize for the typographical error. Herein, we mean compound 14. Appropriate correction was applied.
Reviewer 4 Report
The submitted manuscript reviews the green synthesis of benzimidazole derivatives with pharmacological activity. This review is of great interest from an industrial point of view, suggesting strategies to replace hazardous solvents and reduction of waste production. In addition, this manuscript provides an overview of the pharmacological activities of benzimidazole derivatives. In this regard, the submitted manuscript deals with the current topic and corresponds to the field of the journal.
However, the section on the pharmacological activities is written more comprehensively than the section on the green synthesis (170 references on the pharmacological activity vs. 59 references on the green synthesis). Moreover, almost half of the references related to the green synthesis are self-citations (27 out of 59). Almost all shemes and figures in this section are taken from the original articles of the authors of this manuscript. Therefore, I recommend that the section related to the green synthesis of benzimidazole derivatives be extended with the original works of other authors in order to make the review more complete (major revision).
The proposed minor changes include:
-reformulation of “pharmacological action” into “pharmacological activity”
-reformulation of section 3.3 to “Antiviral activity”
-reformulation of section 3.4. to “Antitumor activity”
-figure 1 should be omitted (not relevant)
-reference 25 should be written correctly.
Author Response
We would like to express our gratitude to the Reviewer for the valuable suggestions in attempt to improve our manuscript. Below are listed the answers to the questions and remarks.
- reformulation of “pharmacological action” into “pharmacological activity”
- Appropriate correction was applied as per Reviewer’s suggestion. Please see revised heading of section 3.
- reformulation of section 3.3 to “Antiviral activity”
- Appropriate correction was applied as per Reviewer’s suggestion. Please see revised heading of section 3.3
- reformulation of section 3.4. to “Antitumor activity”
- Appropriate correction was applied as per Reviewer’s suggestion. Please see revised heading of section 4.
- figure 1 should be omitted (not relevant)
- The reviewer’s suggestion was taken into consideration. Correction was applied, while Figure 1 was removed from the revised manuscript. All figures were renumbered accordingly.
Round 2
Reviewer 1 Report
In my opinion, this manuscript now be accepted in its current form if it fulfils the journal criteria
Author Response
The manuscript meets the journal criteria.
Reviewer 2 Report
The authors fulfilled the revisions requested.
Author Response
In accordance with the Reviewer’s suggestion, we revised the text and the english form.
Reviewer 4 Report
The authors have added four new references to the benzimidazole synthesis section, making it more complete. Minor corrections have also been completed, but it is still necessary to reformulate "pharmacological actions" into "pharmacological activities" in the title of the manuscript. The corrected manuscript contains new lapses that did not exist in the previous paper: the schemes are not numbered well, and reference 67 is missing. Therefore, the new manuscript needs a minor correction.
Author Response
We changed “pharmacological actions” to “pharmacological activities” in the Title of the manuscript. We corrected the scheme numbers and added the reference 67. We reviised the schemes, the text and the english form.